# Seasonal Correction of Offshore Wind Energy Potential due to Air Density: Case of the Iberian Peninsula

**Alain Ulazia [1],\* [ID], Gabriel Ibarra-Berastegi [2,5], Jon Sáenz [3,5] [ID] and Sheila Carreno-Madinabeitia [4] and Santos J. González-Rojí [3,6,7]**

[1] Department of NE and Fluid Mechanics, University of the Basque Country (UPV/EHU), Otaola 29, E-20600 Eibar, Spain

[2] Department of NE and Fluid Mechanics, University of the Basque Country (UPV/EHU), Alda. Urkijo, E-48013 Bilbao, Spain

[3] Department of Applied Physics II, University of the Basque Country (UPV/EHU), B. Sarriena s/n, E-48940 Leioa, Spain

[4] TECNALIA, Parque Tecnológico de Álava, Albert Einstein 28, E-01510 Miñano (Araba/Álava), Spain

[5] Joint Research Unit (UPV/EHU-IEO) Plentziako Itsas Estazioa, University of the Basque Country (UPV/EHU), Areatza Hiribidea 47, E-48620 Plentzia, Spain

[6] Oeschger Centre for Climate Change Research, University of Bern, 3010 Bern, Switzerland

[7] Climate and Environmental Physics, University of Bern, 3010 Bern, Switzerland

\* Correspondence: alain.ulazia@ehu.eus

**Abstract:** A constant value of air density based on its annual average value at a given location is commonly used for the computation of the annual energy production in wind industry. Thus, the correction required in the estimation of daily, monthly or seasonal wind energy production, due to the use of air density, is ordinarily omitted in existing literature. The general method, based on the implementation of the wind speed's Weibull distribution over the power curve of the turbine, omits it if the power curve is not corrected according to the air density of the site. In this study, the seasonal variation of air density was shown to be highly relevant for the computation of offshore wind energy potential around the Iberian Peninsula. If the temperature, pressure, and moisture are taken into account, the wind power density and turbine capacity factor corrections derived from these variations are also significant. In order to demonstrate this, the advanced Weather Research and Forecasting mesoscale Model (WRF) using data assimilation was executed in the study area to obtain a spatial representation of these corrections. According to the results, the wind power density, estimated by taking into account the air density correction, exhibits a difference of 8% between summer and winter, compared with that estimated without the density correction. This implies that seasonal capacity factor estimation corrections of up to 1% in percentage points are necessary for wind turbines mainly for summer and winter, due to air density changes.

**Keywords:** Weather Research and Forecast Model—WRF; data assimilation; air density; offshore wind energy

## 1. Introduction

Energy and environment are strongly related subjects that explain the patterns of the global crisis from the geopolitical viewpoint or from the perspective of climate change. Energy demand is remarkably increasing and this is affecting the environment, not only in a gradual trend, but also with the risk of rapid acceleration due to the permafrost collapse [1]. In 2014, the global primary

energy consumption generated around 10,000 million tons of carbon by the combustion of fossil fuels. Many researchers recognize that renewable energy, especially solar energy and wind energy, can solve these immense challenges [2].

Offshore winds usually present higher speeds than onshore winds, and the available working hours near the rated power of the turbines are also higher offshore. For instance, offshore breezes can be strong almost every afternoon, even matching the time when electrical load demands are highest. Additionally, wind farms cover large areas of land or sea. The area covered by a 3.6 MW turbine can be almost 0.37 km$^2$. As an example, Danish Anholt offshore wind power plant covers a sea area of 88 km$^2$. This is another huge advantage for offshore wind energy, since onshore land occupation of this level presents important ecological and legal problems [3]. However, there is a relevant restriction that is fulfilled in Danish waters: the bathymetry of the location. A maximum depth of 20 m is considered for founded turbines, and of 1000 m for floating turbines and their mooring systems. The authors have developed different offshore wind resource assessment studies in sea areas with different bathymetry such as the Bay of Biscay, Scotland and the Mediterranean taking into account this restriction in the selection of the available area [4–6].

However, there are other important problems against a rapid development of the offshore wind energy industry such as the heavy weight of the nacelle and the corresponding installation costs that can reach the 20% of the total capital cost [7]. The operators also need to wait for adequate and long weather windows to be able to fix damaged turbines in the sea, and this implies the need of the optimization of maintenance routing and scheduling [8]. Typical maintenance cost of an offshore wind turbine is 20% higher than that of an onshore turbine. In this sense, engineers are working in alternative approaches to get a lightweight nacelle for offshore turbines [7] or to measure in-situ the pitch misalignment of the turbine and correct it [9]. These deep and detailed studies on offshore wind energy show the academic, environmental and industrial importance of these renewable source.

That is why we present a novel study for offshore wind energy with a general method to estimate the seasonal energy production variations due to air density. Although the method is applied around the Iberian Peninsula, it can be globally generalized for interesting offshore locations. Air density is a very recent topic of study in wind energy, although its variations are usually considered spatially and in terms of geographical elevation or position [10,11].

In the wind energy industry, wind forecasts and wind energy potential estimation are based mainly on wind speed and only marginally on the temporal and spatial distribution of air density. In wind power literature, the air density is generally considered as constant over time, with a standard value of $\rho_0$ = 1.225 kg/m$^3$ (at sea level, 0 meters above sea level, 0 m.a.s.l., 15 °C). Constant air density and wind speed statistics alone are common starting points in the literature:

- for the estimation of the wind energy potential in specific sites using anemometer data [12–14];
- for the estimation of spatially-distributed wind energy in certain regions using climate data [15] or remote sensing data from satellites [16,17];
- for regional studies by using either reanalysis or mesoscale models [18,19]. In our case, it is carried out by means of the Weather Research and Forecasting Model (WRF) as in the case of Carvalho et al. [20–23]. However, our simulation is a step ahead in the sense that it includes a six-hourly three-dimensional variational data assimilation step which substantially improves its results.

A popular method commonly used for the estimation of *AEP* also omits the air density: first, the wind speed histogram of the studied location is calculated; then, the Weibull distribution is fitted; finally, this distribution is implemented on the turbine's power curve. When there is no power curve correction according to the air density, only the contribution of wind speed is taken into account. This power curve correction can be obviated for the temporal variation of air density as well as for the variation of air density with altitude. Additionally, omitting the variations of air density impacts the

estimation of offshore wind resources because standard air density is defined at 0 m.a.s.l. (1013 hPa), and therefore, it is natural to directly take $\rho_0$ as the actual air density.

In the new context of wind energy industry, the importance of offshore wind energy has increased with the feasibility of implementing floating wind farms, expanding the available area with respect to the limited sites offered by founded turbines [24]. The world's first floating wind farm is operational in Statoil's HyWind Scotland pilot park [25]. Therefore, the massive implementation of these wind farms could be an interesting option for numerous countries [26,27]. Thus, the resource assessment and wind energy potential evaluation must be conducted prior to the technological and economic assessment. We demonstrate that the generally-ignored contribution of the air density's temporal and spatial variations is relevant for the initial step of offshore wind resource evaluation.

There are two main correction methodologies of the wind energy power production due to air density variations:

1.  Obtain a new power curve of the turbine according to the air density of the site. For example:

    1.1.  An important work that takes into account the contribution of air density is authored by Farkas et al. [28]. They analyse the air density's temporal variations (as a function of pressure, temperature, and humidity) in a specific location of Hungary and they propose a correction of the power curve of a specified wind turbine by using neural networks. The air density oscillates around 1.229 kg/m$^3$ (mean value) between 1.395 kg/m$^3$ (high density) and 1.124 kg/m$^3$ (low density). That is, the variations around the mean density are approximately 15%. Since the wind power is proportional to the air density, this is evidently a significant deviation. Apart from that, we conjecture that this is a relevant criterion to evaluate the intra-annual variations. Thus, the target is to establish a reference for the specific location, which will be $\rho_0$ in our case, the value that is commonly used offshore of temperate latitudes such as the Iberian Peninsula.

    1.2.  According to the method of other authors, the correction of the power curve due to air density may involve the complete blade redesigning of wind turbines located at high altitudes, that permanently operate with air densities significantly different from the standard [29,30].

2.  Correct or normalize the wind speed data using the instantaneous air density before the implementation of this time series in the power curve of the turbine. This is our case and there are other examples in the literature:

    2.1.  Collins et al. [31] use a correction method of the wind speed similar to ours for short-forecasting purposes (see Equation (7) and Section 2.2.4). According to their conclusions, for similar wind speed, the difference between the power production on a hot day and that on a cold day can be of the order of 10% for medium wind speeds. This error can be conveniently removed if the forecast contains temperature and pressure data, which can be used to calculate the air density, as in our study.

    2.2.  In Dahmouni et al. [32], an advancement has been made in this regard by introducing better air density information to obtain an effective wind speed (normalized, in our study) in Borj-Cedria (Tunisia). Particularly, the air density has been estimated from observed data of air pressure and air temperature at specified locations in order to normalize the wind speed according to the air density of each instant. Next, the Weibull distribution was fitted, taking into account this new wind speed, and implemented into the power curve. However, the study was developed at specific locations, and the spatial distribution of the wind power correction according to air density was not analysed as in our case.

Regarding to the first methodology, the adjustment of the power curve, it can be performed using the cubic law of wind power. Moreover, rather than scaling the power values directly, which would

alter the rated power of the turbine's power curve ($P_R$), the adjustment is performed by correcting the velocity by means of the following equation recommended in IEC 61400-12 [33]. In this methodology, the site-specific power curve ($U_{site}, \rho_{site}$) and the specified power curve ($U_0, \rho_0$) are related as follows:

$$U_{site} = U_0 \left( \frac{\rho_0}{\rho_{site}} \right)^{1/3} \tag{1}$$

This correction method and ours—described in Section 2.2.5 and given by Equation (7)—are the consequence of the same hypothesis: same power production for the same wind power density. They represent the above mentioned two different ways to obtain the power production, but in an analog manner. Equation (1) is used to create a new power curve of the turbine adapted to the air density of the site, in order to introduce the wind speed directly (without normalization) and obtain the instantaneous power. Equation (7) is used to normalize the wind speed and to insert this normalized value in the original power curve of the turbine which is designed for standard air density. That is why the use of normalization is methodologically more effective and not bounded to the technical constraints of the power curve, since we do not have to change the power curve for each air density in the time series.

Out of the normalization strategy of the wind speed, for the purpose of the introduction of air density fluctuations in the power curve, Svenningsen proposed also another new method that has been adopted by WindPRO [34]: the exponent of the Equation (1) is not constant at 1/3 for all wind speeds, and it is redefined according to three wind speed intervals (0–7, 7–12 and 12–25 m/s). This new method [34] is useful for the wind energy industry in general for the proper correction of the turbine's power curve.

Recently, Eurek et al. [35] have developed a global wind resource estimation method for integrated techno-economic assessment models taking into account, among several other variables, the change in air density with the altitude. They use a straightforward relationship between air density and site altitude [36]. Although air density is a function of both temperature and pressure, which vary by day, season, and geographical location, they use this simplified estimation of air density owing to insufficient temperature and pressure data.

In another recent work, Floors et al. [37] mentioned the importance of air density in wind energy potential estimation, but in their case the subject consists in the reduction of air density at the hub height compared to air density obtained from measurements at lower levels. Additionally, they have found that using re-analysis data (ERA5 in their case) to estimate air density gives similar or smaller errors compared to using nearest measurements around the point of study. A method to interpolate power curves that are valid for site-specific air densities is also presented, but their objective is not to focus on temporal variations and they use annual averages to find the corresponding power curve. In fact, the use of constant site-specific air density is a classical strategy described in well-known works about wind energy resource assessment [38]. These works show energy production variations due to air density similar to energy losses produced by relevant technical aerodynamic or mechanical problems in the turbine [9].

In our case, apart from wind speed, the output from the WRF model presented in [5] yields pressure and temperature at each gridpoint at different $\eta$ vertical levels. From these vertical $\eta$ levels, these fields have been vertically interpolated to geometrical heights above the sea, which have been selected according to the turbine height. Furthermore, Eurek et al. [35] used a global mesoscale model, with a horizontal 40 km spatial resolution, both onshore and offshore, while the run presented here has a 15 km resolution and it fully covers offshore region around the Iberian Peninsula (see Section 2.2.1).

The results of our integration reveal that it is worth considering the variation of the air density during the year in the calculation of seasonal energy production. For that purpose, the WRF model including the data assimilation (WRFDA) is used. The wind speed outputs of this experiment have been previously extensively validated in [5] and compared to ERA-Interim, CCMPv2 satellite analysis and another simulation without data assimilation. Thus, in this case we only validate the density calculated from the virtual temperature and pressure outputs against six buoys around the Iberian

Peninsula (see Section 2.2.3). In this sense, the introduction of the air density in the wind speed time series via a normalization technique is not only used in the annual series which hides the actual seasonal variations due to air density, mainly from winter to summer in middle and high latitudes of the planet. This seasonal analysis due to air density is the main innovation of this work.

The paper is structured as follows: Section 2 describes the data used for the simulations, the verification of the simulations and a reanalysis with the observations. Section 3 presents the main results of the paper. Section 4 discusses the results. Finally, Section 5 presents the conclusions and the outlook.

## 2. Data and Methodology

### 2.1. Data

#### 2.1.1. Buoys

This study was conducted in the period from 1 January 2010 to 1 January 2015. The data used have been obtained from six buoys operated by the Spanish State Ports Authority (Puertos del Estado) [39]. Particularly, temperature and pressure at a height of 3 m.a.s.l. Table 1 presents the position, bathymetry, and distance from the buoys to the nearest gridpoint in the WRFDA integration. Figure 1 shows the map with the six buoys around the study area and the domain covered by the WRFDA simulation.

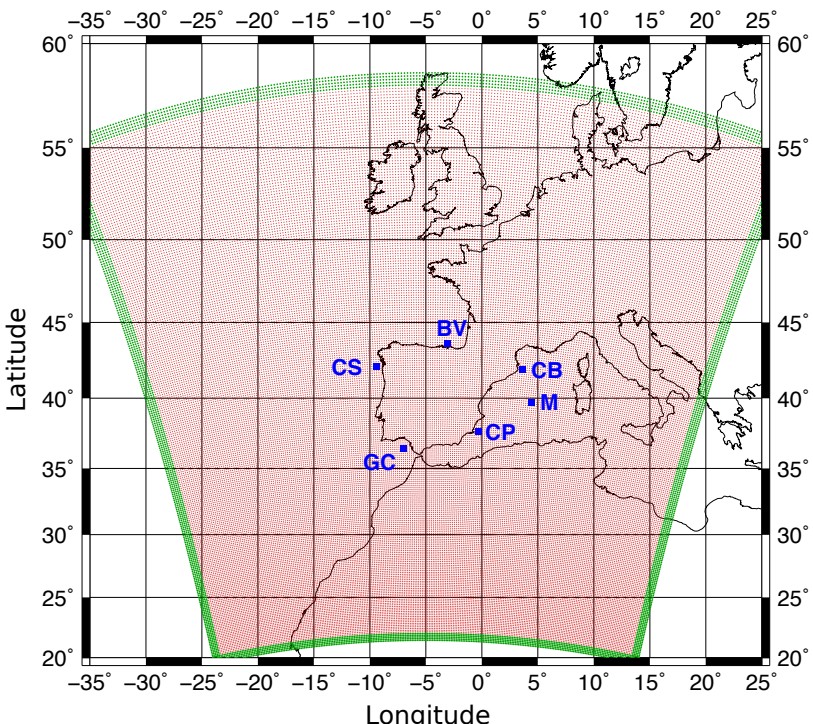

**Figure 1.** WRFDA simulation domain and the positions of the six buoys.

**Table 1.** Properties of the buoys and stations [39].

| Name | Long. (°E) | Lat. (°N) | Depth (m) | Dist. (km) |
|---|---|---|---|---|
| Cabo de Palos, CP | −0.30 | 37.65 | 230 | 6.7 |
| Cabo Begur, CB | 3.65 | 41.92 | 1200 | 3.7 |
| Mahón, M | 4.42 | 39.72 | 1200 | 4.1 |
| Bilbao Vizcaya, BV | −3.05 | 43.64 | 600 | 3.2 |
| Golfo de Cádiz, GC | −6.96 | 36.48 | 300 | 5.0 |
| Cabo Silleiro, CS | −9.43 | 42.12 | 615 | 6.2 |

### 2.1.2. Wind Turbine

The selected wind turbine is the MWT-92/2.4 (2.4 MW rated power and 92 m diameter) Mitsubishi Heavy Industry's turbine, that has been studied in Japan waters ([40], [p. 511]). Thus, the results in this study are computed based on real wind energy technology. Its power curve with a wind speed interval of 0.5 m/s has been used for the estimations of the energy production errors due to air density. Figure 2 shows this curve and the main characteristics of the turbine [41].

**Figure 2.** Power curve and characteristics of the MWT-92/2.4 turbine [40].

### 2.2. Methodology

### 2.2.1. WRF Simulation

A six-year-long (2009–2014) experiment was created using version 3.6.1 of the WRF model [42] and the WRFDA data assimilation [43–45]. The experiment was started on 1 January 2009; however, that year was used only as a spin-up for the soil system in the model, and it is not included in this paper as is typically done in other literature [46,47]. The WRF model was nested inside ERA-Interim [48]. This reanalysis provides the initial and boundary conditions for the execution, at 0.75 degrees spatial resolution and 20 vertical levels up to 5 hPa. In this experiment, six-hourly boundary conditions drove the model after the initialization from a cold start; the 3DVAR data assimilation [45] step was also executed on it every 6 h (at 00, 06, 12 and 18 UTC). To achieve this, observations from NCEP ADP Global Upper Air and Surface Observations dataset were retrieved from NCAR/UCAR's Research Data Archive (referenced as ds337.0).

The domain focuses on the Iberian Peninsula (Figure 1); north-western Africa and western Europe are also covered. According to [49,50], the domain is large enough to prevent border effects over the studied region, and mesoscale meteorological features can develop freely inside the domain. The experiment was defined at 15 km grid resolution and 51 vertical levels. This horizontal resolution is more accurate than that from ERA-Interim, providing a better representation of the topography of the Iberian Peninsula.

As stated before, ERA-Interim Reanalysis data downloaded from the Meteorological Archival and Retrieval System (MARS) at ECMWF were used as the boundary/initial conditions of the simulation. The SST used as the lower boundary condition by WRF was updated daily using the NOAA OI SST V2 High Resolution Dataset [51]. Finally, the model's physical parametrization schemes used were as follows: five-class microphysics scheme (WSM5), longwave and shortwave radiation scheme (RRTMG), planetary boundary layer scheme (MYNN2), Tiedtke cumulus parameterization, and NOAH land surface model.

The background error covariance (BE) matrices used in the 3DVAR analysis step were calculated by means of the CV5 option in WRFDA [52]. A BE matrix exists for each month that can only be used on this simulation as it takes into account the domain of the simulation and the parameterizations used. The BE matrices were created by means of a special 13-month execution (starting on 1 January 2007), initialized at 00 and 12 UTC. Each matrix was created from 90 days of this special integration; that is, in order to create February's matrix, data from January, February, and March were required; similarly, for all months.

The WRFDA experiment's output is composed of 12-h long segments, starting from each analysis time (00, 06, 12 and 18 UTC). The analyses are produced by using the previous segment's output at a 6-h forecast as the first estimate for the 3DVAR data assimilation step. The outputs were stored every 3 h so that the analyses (00, 06, 12 and 18 UTC) and three-hour forecasts (03, 09, 15 and 21 UTC) were saved. Table 2 shows the main characteristics of the simulation.

The raw model outputs were post-processed. The pressure, virtual temperature of the air and wind speed were estimated at different turbine heights. They were interpolated from the results at the original vertical $\eta$ levels of the model. Thereby, the data in the levels interpolated at 70 m are used in this case to obtain the wind speed at the hub height of the MWT-92/2.4, in the power curve of which the wind data will be implemented as explained in the Section 2.2.5.

A previous study carried out by the authors [53] demonstrated that this simulation, including data assimilation, is effective for improving the results provided by the numerical integrations where only the boundary conditions drive the model, for variables such as the precipitation, evaporation and precipitable water over the Iberian Peninsula. Additionally, its results are superior (or at least comparable) to those from the ERA-Interim reanalysis. As mentioned before, identical conclusions were obtained for the wind field simulated over the West Mediterranean [5], where WRFDA outperformed the conventional numerical integration and the driving reanalysis, producing more realistic winds.

**Table 2.** Main characteristics of the WRFDA experiment.

| | |
|---|---|
| Spatial resolution | 15 km × 15 km |
| Vertical levels | 51 $\eta$ levels up to 20 hPa |
| Temporal resolution | Outputs stored every 3 h |
| Parametrizations used | Five-class microphysics WSM5<br>RRTMG (longwave and shortwave) radiation<br>Mellor-Yamada Nakanishi and Niino Level 2.5 PBL<br>Revised MM5 surface layer scheme<br>Tiedtke cumulus parameterization |
| Data used | PREPBUFR from the NCEP ADP Global Observations for data assimilation<br>Boundary conditions from ERA-Interim fields (ECMWF) and NOOA OI SSTV2 |

### 2.2.2. Dry Temperature and Virtual Temperature

As the buoys yield temperature and pressure data and the moisture is not measured, air density has been computed using the Equation (4), albeit by considering the dry temperature (T) rather than the virtual temperature ($T_v$). It is known that the density of dry air is higher than that of humid air because water's molecular weight is lower than those of the most abundant $N_2$ or $O_2$ species that constitute dry air. Thus, there should be an underestimation in the values of air density by WRFDA with respect to the values derived from the buoys, as shown by the results (see Section 3.2).

The contribution of moisture to the air density is negligible as it is widely stated in existing literature [31,54]. However, we have studied this influence using the package *aiRthermo* created in the R programming language [55,56]. This package addresses numerous computations related to the thermodynamics of the atmosphere and includes a number of functions designed to consider the density of air with varying degrees of water vapour in it.

Since the amount of moisture in the atmosphere is highly variable in time and space, the mixture of gases in the atmosphere is commonly considered as dry air with constant composition and water vapour [57]. Virtual temperature is the temperature that dry air should have in order that its density is equal to moist air's density. Thus, $\rho_m = \rho_d + \rho_w$ with $\rho_m$, $\rho_d$ and $\rho_w$ the densities corresponding to moist, dry air and water vapour, respectively. Considering Dalton's law for pressures and the perfect gas state equation for air and water vapour, we get from the previous equation

$$\frac{P}{R_d T_v} = \frac{P - e}{R_d T} + \frac{e}{R_v T} \tag{2}$$

with $P$ atmospheric pressure, $T$ air temperature, $T_v$ virtual temperature, $e$ partial pressure by water vapour and $R_d$ and $R_v$ the gas constants for the dry air and water vapour, respectively. From this expression, we can arrive to the equation that yields the value of virtual temperature [58]

$$T_v = \frac{T}{1 - (1 - \varepsilon) \frac{e}{P}} \tag{3}$$

with $T$ temperature, $P$ pressure, $e$ partial pressure of water vapour and $\varepsilon = \frac{R_d}{R_v}$ the ratio of the gas constants for dry air and water vapour, respectively [55].

For example, we have compared $T$ and $T_v$ using *aiRthermo* for the case of the BV buoy. In Figure 3, the temporal evolution of the difference in percent between both temperatures is shown for all the cases registered by the buoy under two extreme cases of relative humidity: 30% (black) and 100% (red). In the worst cases, the deviation reaches approximately 1.3%, but the average is around 0.7%. Therefore, although this contribution is small, our WRFDA model presents the advantage of considering its effect also, which is not captured by the buoy data.

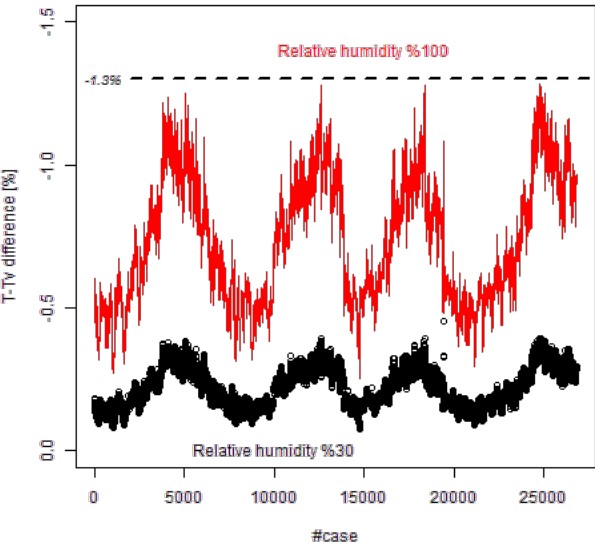

**Figure 3.** Differences (in percent) between $T$ and $T_v$ at BV buoy for two extreme values of moisture.

### 2.2.3. WRDFA Model Results Verification with Experimental Data

The authors would like to stress that the paper deals with seasonal changes in the air density (Figures 4 and 5), the close agreement between density calculated from model and observations (Figure 6, Figure 7 and Table 3), and the consequent results on Wind Power Density (Figure 8) and seasonal capacity factors for winter and summer (Figure 9).

**Table 3.** R-squared, *p*-value and Slope of the model versus the buoys for pressure (left) and temperature (right).

| Buoy | R-sq. (P) | *p*-val. (P) | Slope (P) | R-sq. (T) | *p*-val. (T) | Slope (T) |
|------|-----------|--------------|-----------|-----------|--------------|-----------|
| BV | 0.99 | $2.2 \times 10^{-16}$ | 0.94 | 0.95 | $2.2 \times 10^{-16}$ | 0.94 |
| CB | 0.97 | $2.2 \times 10^{-16}$ | 0.94 | 0.97 | $2.2 \times 10^{-16}$ | 1.00 |
| CP | 0.98 | $2.2 \times 10^{-16}$ | 0.95 | 0.97 | $2.2 \times 10^{-16}$ | 0.98 |
| CS | 0.99 | $2.2 \times 10^{-16}$ | 0.96 | 0.91 | $2.2 \times 10^{-16}$ | 0.98 |
| GC | 0.97 | $2.2 \times 10^{-16}$ | 0.96 | 0.93 | $2.2 \times 10^{-16}$ | 1.01 |
| M | 0.98 | $2.2 \times 10^{-16}$ | 0.93 | 0.92 | $2.2 \times 10^{-16}$ | 0.93 |

The air density outputs from the pressure and temperature values (Equation (4)) at 10 m.a.s.l. at approximately 2500 gridpoints cover the area corresponding to the WRFDA simulation. They have been evaluated with respect to the buoy's observations every 3 h. The evaluated period is from 2010 to 2014.

Three statistical indicators have been used for that evaluation:

1. Pearson's correlation coefficient ($r$),
2. the root mean square error ($RMSE$),
3. and the standard deviation ratio ($SD$ ratio) between the $SD$ of the model and the $SD$ of the observation.

These verification indices are represented by the Taylor diagram [59], which is commonly used for the analysis of model errors. The correlation is represented by the exterior arc of the diagram and the $RMSE$ by the arc centred in the observation point; moreover, the $SD$ ratio's value is one for the interior arc that passes through the observation.

In all cases, the differences in performance according to this set of indicators have been assessed at a 95% confidence level with 1000 samples created by means of the bootstrap technique with resampling, and representing it by a cloud of points in the corresponding Taylor diagrams. A preliminary data-pre-processing was carried out to arrange the data from the buoys and model along a common time line. A total of 14608 pairs observation-simulation were used.

Another evaluation has been also developed using boxplots to compare the air density percentiles of each season between the WRFDA model and the buoy observation. That is, the 25%, 50% and 75% percentiles in the limits and in the middle (the median) of the boxes—and extreme events by means of the whiskers and exterior points.

Furthermore, the Mean Absolute Percentage Error ($MAPE$) has been computed for each season and each buoy. It is a common statistical index in wind energy estimation and forecasting [60–62] or even in wave energy studies [63]. $MAPE$ uses natural logarithmic values of the models and observations rather than raw values. Thereby, the range differences can be rescaled, and a homogeneous comparison based on the proportional error in percent is obtained.

Finally, temperature and pressure have been separately evaluated, since the moisture has been already evaluated in González-Rojí et al. [53]. In this way, we obtain an independent validation for each essential meteorological variable involved in the final value of air density. R-squared, *p*-value and the slopes of the model versus the buoys have been obtained by means of scatterplots.

### 2.2.4. Wind Power Density Calculation

After the validation of our simulation, the corresponding outputs have been used to calculate the seasonal capacity factor (*SCF*) and seasonal wind power density (SWPD) for the mentioned specific

offshore floating turbine: MWT-92/2.4. As explained before, an important advantage of our study is that the pressure, temperature, and wind values from the simulation were directly interpolated from the original $\eta$-levels in the WRF/WRFDA system to the hub height.

As wind power is proportional to both the density and to the cube of the wind speed, we have used the magnitudes of the virtual temperature ($T_v$, in Kelvins) and pressure ($P$, in Pascals) of the WRFDA simulation to obtain the density $\rho$ of the air (in kg/m$^3$) at the hub height as in Section 2.2.2

$$\rho = \frac{P}{R_d T_v} \tag{4}$$

So the computed air density is accurate as additional data about the local humidity are included via the virtual temperature.

Thus, we can finally define the wind power density (*WPD*) according to the air density and wind speed in a time-series of N cases $i\epsilon\ \{1, ..., N\}$:

$$WPD = \frac{1}{2} \sum_{i=1}^{N} \rho_i U_i^3 \tag{5}$$

and, therefore, the relative error in percent (*WPDr*) between the *WPD* that takes into account the density variations and the *WPD* that does not consider them is as follows:

$$WPDr = \left( \frac{\sum_{i=1}^{N} \rho_i U_i^3}{\rho_0 \sum_{i=1}^{N} U_i^3} - 1 \right) \times 100 \tag{6}$$

$\rho_0$ being the standard air density that is commonly used. This relative error offers a measure of the contribution of air density to the *WPD*; this is mainly so if we consider the time series of the four seasons versus the wind power yielded by the standard air density.

### 2.2.5. Capacity Factor Calculation

A normalized wind speed has been calculated to incorporate the influence of air density into the power curve of the MWT-92/2.4. Figure 4 illustrates this implementation with the power curve (upper panel), and the normalized and non-normalized wind speed distributions (lower panel). Note that the horizontal axis of wind speeds matches above and below. Discrete histograms are used in our computation, but as it is shown, the use of fitted Weibull distributions would be equivalent. However, the two Weibull curves show clearly the difference between the normalized and non-normalized distributions, and they are visualized here for explanatory purposes. As it can be seen, the difference is quantified in the value of the $c$ scale parameter: 5.11 for the normalized distribution and 5.47 for the non-normalized. These distributions correspond to the nearest grid point to Cabo de Palos in a summer with high temperatures. As it is mentioned below, more histogram bins have been used in the actual computation, but we used less and wider bins here to show an illustrative idea of our methodology.

The normalized wind speed $U_n$ is defined first for each case of the wind speed and air density time series:

$$U_n = U \left( \frac{\rho}{\rho_0} \right)^{1/3} \tag{7}$$

Then we have obtained the separated time series of each season to construct the histograms of normalized wind speed with an interval of 0.5 m/s from 0 to 40 m/s. The same interval has been used in the power curve of our turbine considering zero KW above the cut-off value (25 m/s). In this way, wind statistics can be implemented in the power curve obtaining seasonal energy production (*SEP*). The quantity of cases in each interval is given by the discrete probability density $f_i$ (which fulfills $\sum_{i=1}^{80} f_i = 1$) for each interval $i$, and $P_i$ is the mean power at that interval $i$ in MW:

$$SEP(TWh) = \sum_{i=1}^{80}(f_i P_i) \cdot (365.25/4) \times 24 \times 10^{-6} \tag{8}$$

Thus, we have the seasonal capacity factor *SCF* restricted to the time-series of a specified season, for example, winter, being $P_R$ the rated power of the turbine in MW (2.4 MW):

$$SCF_{winter} = \frac{SEP_{winter}}{P_R \cdot (365.25/4) \times 24 \times 10^{-6}} \tag{9}$$

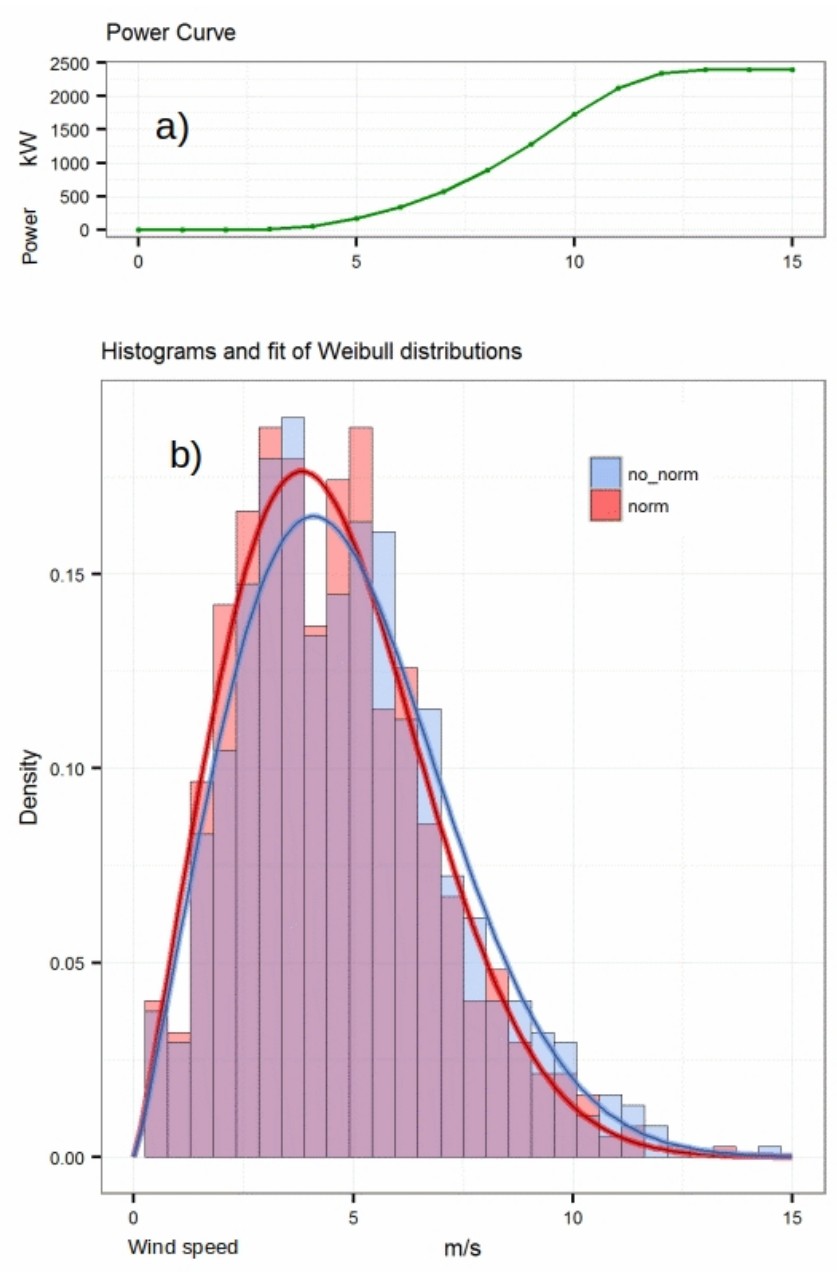

**Figure 4.** Illustration of the implementation of the wind speed distributions on the power curve of the turbine. Above (**a**), the power curve of the MWT-92/2.4 turbine is matched for the same wind speeds. Below (**b**), histograms of normalized and non-normalized wind speeds and corresponding fit of Weibull distributions during a summer with high temperatures in Cabo de Palos. Parameters: $k = 2.2$ in both cases; $c = 5.1$ (normalized distribution) and 5.5 (non-normalized).

## 3. Results

### 3.1. Monthly Air Density Percentile at the Buoys

The results in Figure 5 exhibit evident seasonality in the value of the air density, decreasing gradually from winter to summer. The percentile boxes of each month also reveal that in winter (J, F and M), the monthly air density variations are stronger than those in the other seasons; those in summer (J, A and S) variations are negligible. In the strongest case (BV), the median can change from 1.25·kg/m$^3$ in January to 1.20 kg/m$^3$ in August, which implies a relative variation of over 4% around the standard air density ($\rho_0$) of 1.225 kg/m$^3$. The other cases are similar, with the lowest winter–summer relative variation of 2% at CS. These representations also reveal that the intra-monthly variations are low compared to seasonal changes and that the seasonality must therefore be mainly considered. There are months and sites (for example February at BV) where the extreme cases within the month (from 5th to 95th percentile) undergo an absolute change of over 0.1 kg/m$^3$, that is, approximately 9% around $\rho_0$.

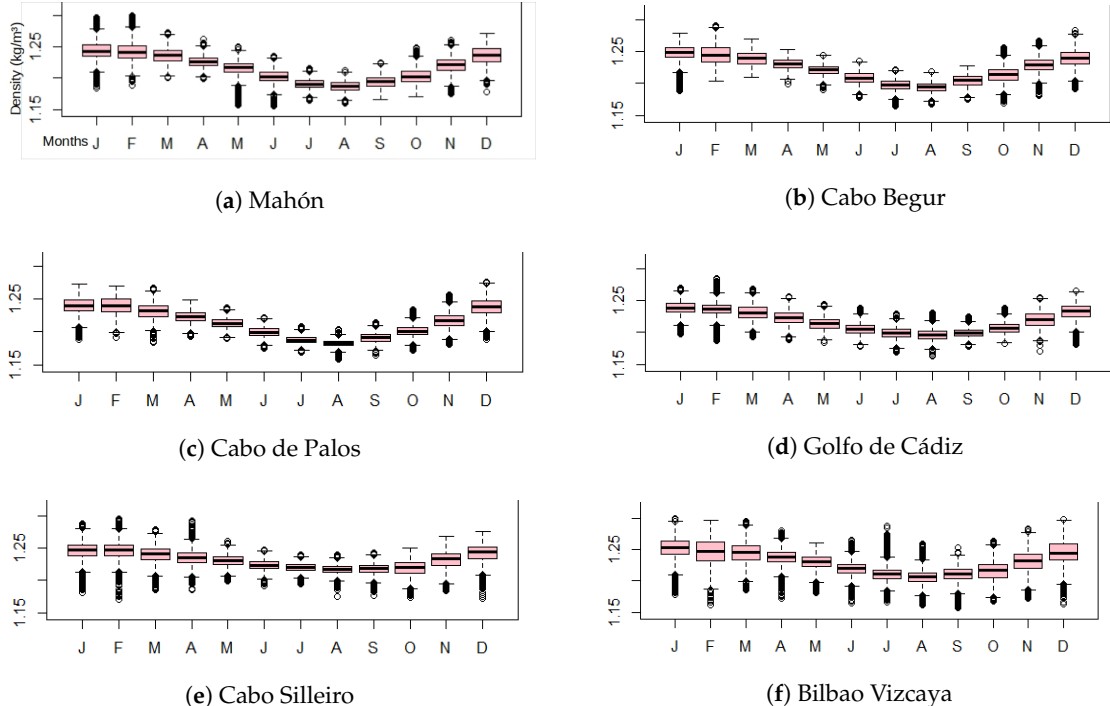

**Figure 5.** Monthly quantile boxplots of air density at the six buoys with variations that can reach almost the 10% around the standard air density (in February at BV): (**a**) Mahon, (**b**) Cabo Begur, (**c**) Cabo de Palos, (**d**) Golfo de Cadiz, (**e**) Cabo Silleiro, (**f**) Bilbao Vizcaya.

Owing to the higher summer temperatures in the Mediterranean Sea, the air density in summer is lower than that in the Atlantic coast. This is consistent with the classical climate characterization for the Iberian Peninsula [64].

### 3.2. Verification versus Buoys

Figure 6 shows the Taylor diagrams against the six buoys. Considering our three statistical indicators represented in the diagrams, the validation results are very good:

- The correlations represented by the exterior arc are over $r = 0.9$ in the six cases, reaching 0.95 in certain cases. Considering that the number of elements in our samples is in all cases longer than 11,000 (taking into account the NA values of the buoys), the correlation coefficients are significant to 99% according to a two-tailed t-test.

- *RMSE*, represented by the interior arc centred in the observation point, is below 0.01 in each case with a limit value of 0.005. Therefore, it implies a relative error between 1% and 0.5%.
- *SD* ratio, which is equal to one over the interior arc parallel to the correlation arc, is between 0.85 and 1.10 in the six cases.

These results indicate that the densities calculated with the WRFDA model exhibit adequate agreement with observations at six representative locations. Thus, it verifies that the WRFDA outputs with a 15 × 15 km resolution can be used to calculate and describe the density variations along the coast as in our previous case [5].

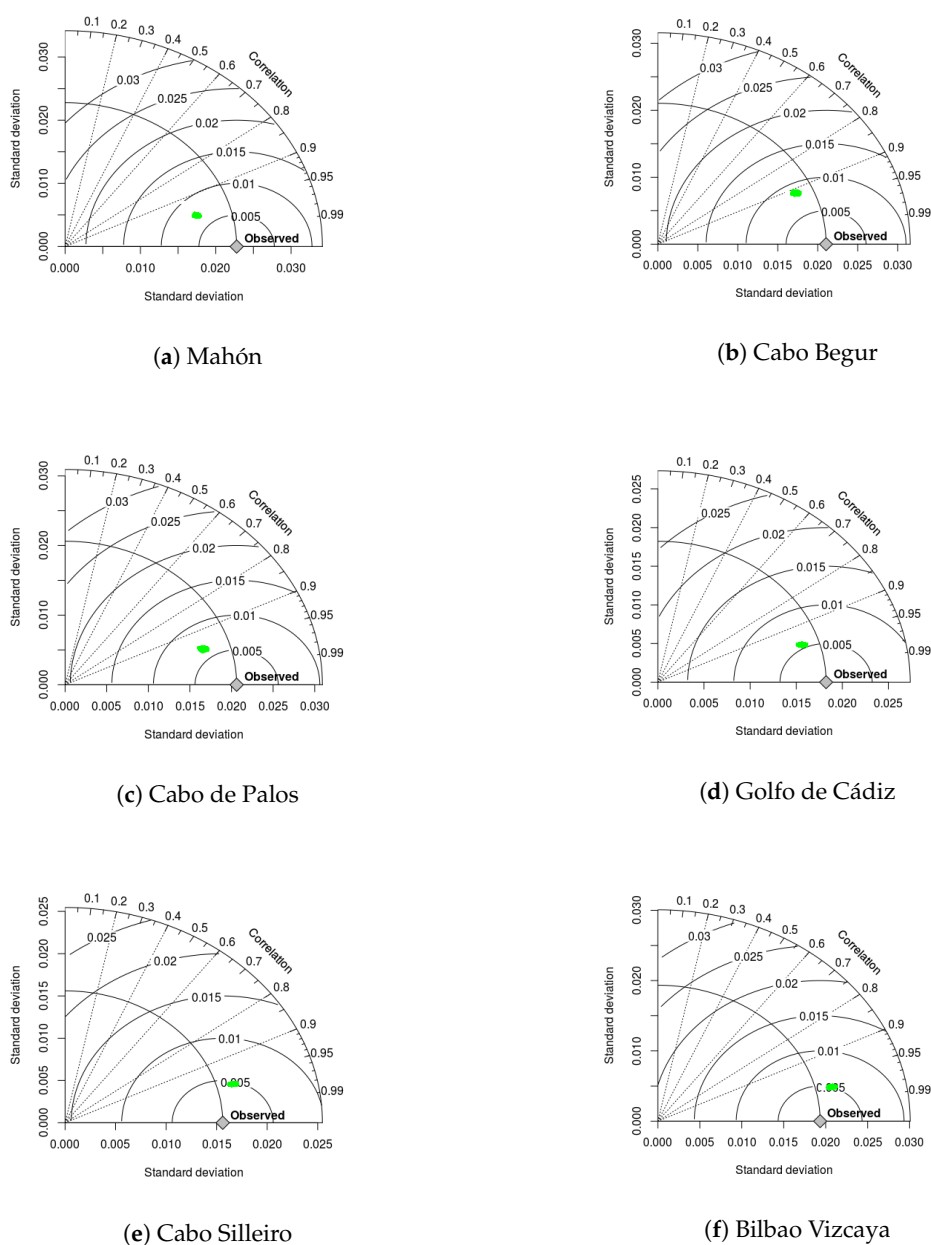

(**a**) Mahón

(**b**) Cabo Begur

(**c**) Cabo de Palos

(**d**) Golfo de Cádiz

(**e**) Cabo Silleiro

(**f**) Bilbao Vizcaya

**Figure 6.** Taylor diagrams for air density at the six buoys: (**a**) Mahon, (**b**) Cabo Begur, (**c**) Cabo de Palos, (**d**) Golfo de Cadiz, (**e**) Cabo Silleiro, (**f**) Bilbao Vizcaya. The green dots represent the position of the model-simulated air density according to the observed one, in terms of correlation coefficient (angle), standard deviation (radius) and root mean squared error [59].

In Figure 7, seasonal boxplots and *MAPE* values of WRFDA and the observation are shown for each buoy. The *MAPE* values are below 1% in each case. This implies that the errors in the computation of *CF* will be smaller because it is related to the 1/3rd power of the air density (1/3 of air density's relative error according to error theory). The boxplots exhibit highly similar seasonal trends, with a marginal general underestimation of the model with respect to the observation, with the exception of Cabo de Palos. However, these deviations of the model's boxplots are approximately $\pm 0.01$ kg/m$^3$, which is coherent with *MAPE* values below 1%.

In addition to all this validation for air density, pressure and temperature have also been validated separately, since moisture has already been evaluated in [53]. To do so, the nearest WRFDA grid cells to the buoys have been defined, and the scatterplots of WRFDA versus buoy data were plotted. A linear regression model between simulated and observed values was also computed. These plots are not shown here, but the R-squared, *p*-value and slope of each buoy from the linear models for pressure (left) and temperature (right) are summarized in Table 3. The R-squared and slopes are very close to 1, and the *p*-values are much lower than 0.01. These results highlight WRFDA's ability to correctly simulate the pressure and temperature near the Iberian coasts.

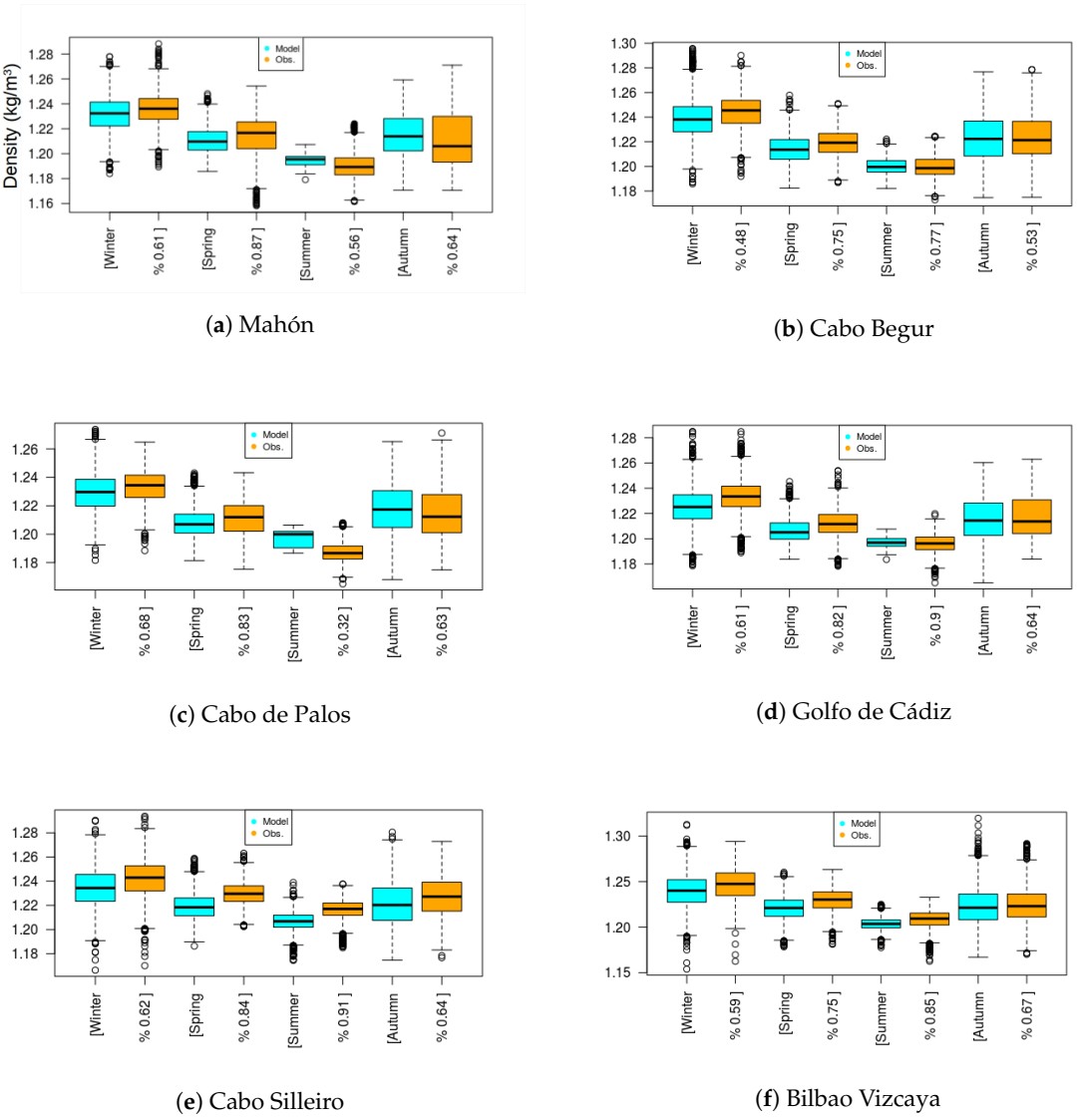

(**a**) Mahón

(**b**) Cabo Begur

(**c**) Cabo de Palos

(**d**) Golfo de Cádiz

(**e**) Cabo Silleiro

(**f**) Bilbao Vizcaya

**Figure 7.** Seasonal boxplots and $MAPE = |\log(model) - \log(observation)| \times 100$ at each buoy. (**a**) Mahon, (**b**) Cabo Begur, (**c**) Cabo de Palos, (**d**) Golfo de Cadiz, (**e**) Cabo Silleiro, (**f**) Bilbao Vizcaya.

### 3.3. Seasonal Wind Power Density Maps

The relative error in percent at each gridpoint between the *WPD* calculated taken into account the time-series of the air density and the *WPD* calculated considering a constant standard air density (*WPDr*) exhibits relevant values in our maps, mainly in winter and summer; this demonstrates that the deviation can reach −6% around Gulf of Cádiz in summer. A divergent color bar centered on zero is used to keep the visual symmetry between negative and positive values. The winter map reveals that the relative error is not that critical because $\rho_0$ is approximately equal to the mean air density in winter in these latitudes. However, the error can be +2% in the east of Bay of Biscay and Cabo Begur. Spring and autumn represent a smooth transition between winter and summer.

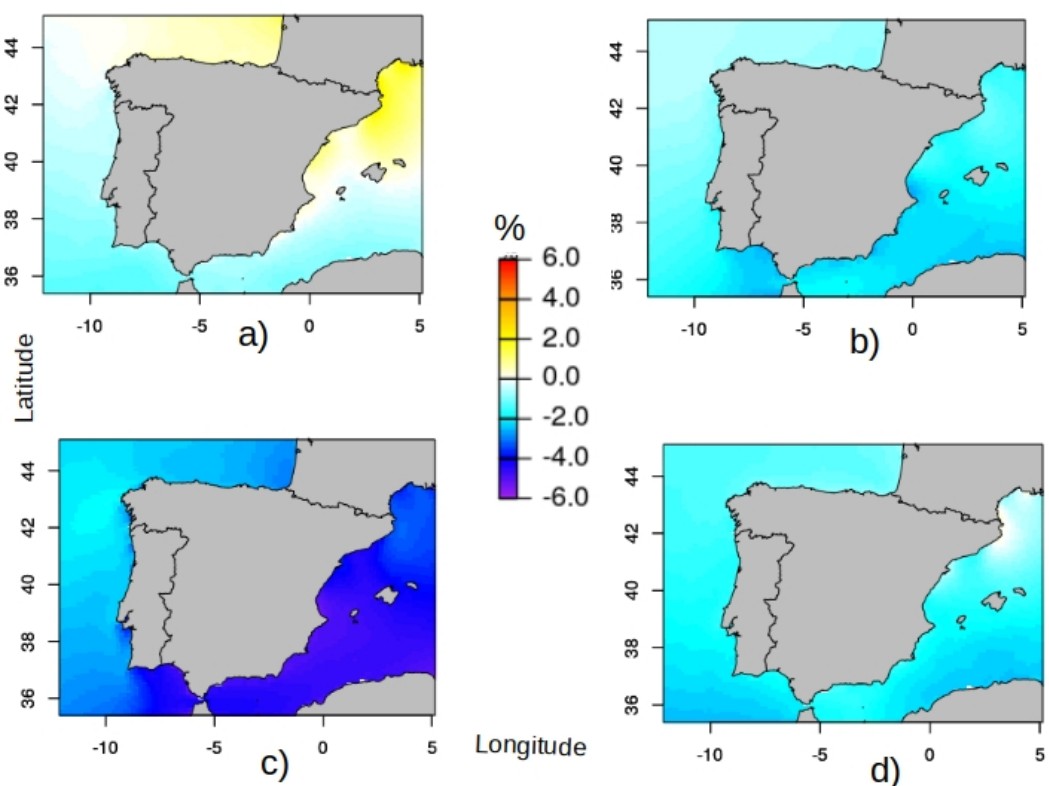

**Figure 8.** Wind Power Density relative errors for the four seasons in the study area: (**a**) Winter, (**b**) Spring, (**c**) Summer, (**d**) Autumn

### 3.4. Seasonal Capacity Factor Maps: Winter and Summer

As the main variations in *WPD* are exhibited in winter and summer, we have represented only the two *SCF* maps for these seasons (see Figure 9). The figure shows the error in percentage points at each gridpoint of our simulation between the seasonal *CF* calculated considering the seasonal contribution of the air density and that without this consideration (Equation (6)). There are substantial differences between winter and summer in areas such as Cabo Begur: more than 1.4% seasonal error variation, from +0.4% in winter to −1% in summer. The Gulf of Cádiz is also significant: 0.9% error variation, from −0.3% in winter to −1.2% in summer. Moreover, in the largest portion of the shore of the Iberian Peninsula, the correction owing to the consideration of the air density time-series reaches 0.5%.

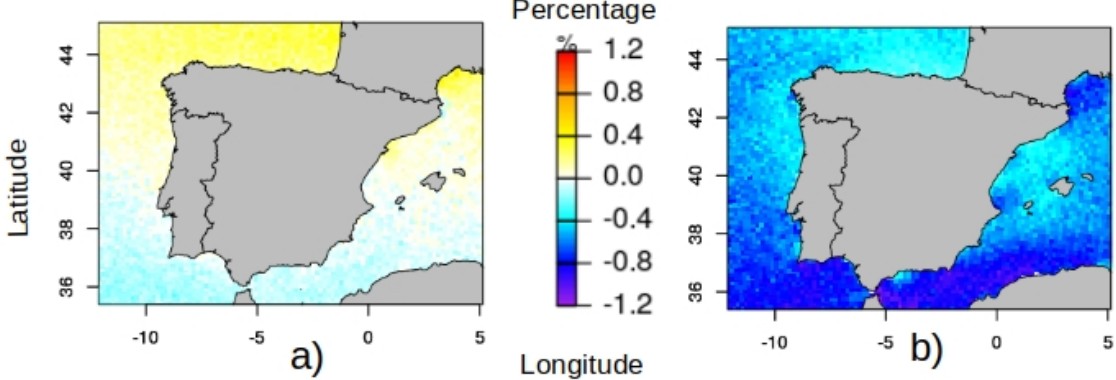

**Figure 9.** Maps representing the error in percentage terms between the seasonal *CF* of winter (**a**) and summer (**b**) owing to the consideration of air density

## 4. Discussion

A correction of $\pm$ 1% in *SCF* owing to the consideration of air density implies a *SEP* correction of $\pm$ 0.22 GWh for our turbine of 2.4 MW (Equation (8)); that is, more than 40,000 US$ seasonally per turbine if we consider the 2017 price of 1 kWh in Spain [65]. Alternatively, assuming a typical *COE* of 0.06 US$/kWh in wind energy [66], this correction implies a capacity of initial inversion of approximately $\pm$8000 US$ for one turbine. 1% is a referential and generalist value in our study, although we have shown that the oscillations from winter to summer can be even higher at some places. Moreover, it is obvious that these economical values are approximately proportional to the rated power of the wind turbine; that is, a turbine of 10 MW would imply a *SEP* correction of almost 1 GWh.

The relevance of these values is indubitable if they are compared with other kind of losses in wind energy related to important technical problems. For instance, the authors have studied the effect of pitch misalignment and defective anemometers in wind turbines, and the order of magnitude of energy losses is quite similar [9,67]. This comparative fact emphasizes the economical relevance of seasonal energy variations due to air density changes.

As is evident, the impact of density is not uniform along the coasts of a relatively small area such as the Iberian Peninsula, and regionalized estimations are required. This indicates that if this methodology is to be applied in other regions, it is likely that regional models such as WRFDA with a rather high resolution may be necessary to effectively characterize the spatial variability of the impact that density has on wind energy production.

It is also important to highlight that as demonstrated by the authors in previous works [5,6] related to the calculation of wind energy from meteorological models such as WRF, there is a significant improvement in the accuracy of the estimations if the model is executed with data assimilation (WRFDA). Therefore, the only WRF configuration considered in this paper has been the one including data assimilation. The high consistency with local measurements at six buoys validates this.

This study has been carried out for a five-years period; therefore, the results depict a significant average economic impact per turbine associated to the 2010–2014 period. However, wind turbines are expected to operate for periods of 20–30 years or beyond. This indicates that a realistic economic feasibility analysis of future wind farms need to incorporate the current seasonal variations of air density and wind speed as well as future estimations, as provided by AR5/CMIP5 models (https: //esgf-node.llnl.gov/search/cmip5/). Similarly, currently operating wind farms may also require an economic re-evaluation in order to incorporate the economic impact of the inter-seasonal variability of density as well as the long-term trends that may already be affecting the variables involved (temperature, pressure and wind). A recent study by the authors [68] on the wave field closely interconnected to the wind field in the Bay of Biscay clearly indicates that a positive upward trend

is occurring in this area at least since the early 20th century. Although not directly applicable to temperature, pressure and wind, these results at least indicate that this issue must be addressed to elucidate whether these trends quantitatively affect the evolution of the *CF* in the future.

The results are completely consistent with fundamental features of the climatology of the Iberian Peninsula. Summer months (JJA) are characterized by approximately 10 K warmer surface temperatures over the Iberian Peninsula than those in winter months (DJF). This results in lower air densities over the area. Additionally, higher temperatures together with the Clausius–Clapeyron equation result in higher saturation mixing ratios over offshore areas that also tend to produce lower densities over the area. Finally, the seasonal cycle of surface pressures results in lower (approximately 5 hPa) mean sea level pressure over the area during summer than during winter. Thus, the seasonal cycle in surface pressure also produces a lower density over the area. Therefore, the results presented in section 3 are consistent with known features of the seasonal cycle of temperature, mixing ratio and sea level pressure over the area.

Given that the main purpose of this paper is to emphasize the importance of the contribution of air density to the seasonality of wind energy production, the physical explanation of the changes in the air density owing to pressure, temperature and moisture is not analysed here in detail, although they are highly popular, taking into account the atmospheric thermodynamics relating pressure, temperature, moisture, and density [57,69,70].

## 5. Conclusions

A strong spatial variability of wind energy due to air density changes has been detected offshore around the Iberian Peninsula, and the need for a regionalized approach arises. This implies that a meteorological model such as WRF that operates at a rather high resolution is required to characterize the spatial variability detected. Using WRF with data assimilation provides an accurate representation, as established by previous studies and the high consistency with density observations from the buoys. The methodology presented in this paper is not site-dependent and can be applied to all regions worldwide.

Further research is currently being carried out by authors in order to incorporate the effects of climatic variability into these conclusions to the same areas of the Iberian Peninsula. AR5/CMIP5 models routinely provide monthly outputs of temperature, pressure and wind. After a proper selection of the best model(s) for these target variables and areas, the methodology explained in this work will be used for a more accurate life-time overall evaluation of wind farms.

The WRF simulation using data assimilation used for this study has also been used for another offshore wind energy study over the Mediterranean at different hub heights [5]. Given the great hub heights of modern turbines, these mesoscale models are interesting, because they give a good approximation of wind speed at these heights when vertical interpolation from the original $\eta$ model levels is used. Anyway, Floors et al. [37] have shown that air density should be also corrected. For instance, an air density reduction of 2% is expected according to their model for a turbine with a tower of 200 m. Therefore, in future studies about seasonal variations of air density this aspect of the hub height and the consequent reduction of air density should be introduced. Additionally, the reduction rate of temperature with altitude in high latitudes (in middle latitudes is 0.0065 K/m according to the US standard atmospheric model [71]) shows a very special behavior, and this kind of anomalies should change the correction models of air density due to the hub height in Arctic conditions, where, paradoxically, the temperature changes from winter to summer (therefore, air density changes) are also strong. Deserts can also be of interest, not for the seasonal study, but for the daily analysis, since there are strong temperature oscillations from day to night.

Moreover, new reanalysis outputs as ERA5 can be used in the future to develop similar studies in an European or even global scale, since it has recently presented very good results for wind energy estimation [72]. In this line, a recent study of the authors has shown great variations due to air density from winter to summer in Hywind-Scotland pilot wind farm using ERA5 [4]. The authors are

developing further research to study also daily variations and extreme cases that can be related to short-term wind forecasting, which constitutes a relevant research line for wind-related industry [73].

**Author Contributions:** Conceptualization, A.U.; Methodology, A.U.; Software, A.U., J.S., S.J.G.-R.; Investigation, A.U., G.I.-B., J.S., S.C.-M.; Writing, Review & Editing, all the authors; Supervision, all the authors; Project Administration, G.I.-B.; Funding Acquisition, G.I.-B. and J.S.

**Funding:**　This work has been funded by the Spanish Government's MINECO project CGL2016-76561-R (AEI/FEDER EU) and the University of the Basque Country (UPV/EHU funded project GIU17/02). The ECMWF ERA-Interim data used in this study have been obtained from the ECMWF-MARS Data Server. The authors wish to express their gratitude to the Spanish Port Authorities (Puertos del Estado) for being kind enough to provide data for this study. The computational resources used in the project were provided by I2BASQUE. The authors thank the creators of the WRF/ARW and WRFDA systems for making them freely available to the community. NOAA_OI_SST_V2 data provided by the NOAA/OAR/ESRL PSD, Boulder, Colorado, USA, through their web-site at http://www.esrl.noaa.gov/psd/ were used in this paper. National Centres for Environmental Prediction/National Weather Service/NOAA/U.S. Department of Commerce. 2008, updated daily. NCEP ADP Global Upper Air and Surface Weather Observations (PREPBUFR format), May 1997—continuing. Research Data Archive at the National Centre for Atmospheric Research, Computational and Information Systems Laboratory. http://rda.ucar.edu/datasets/ds337.0/ were used. All the calculations have been carried out in the framework of R Core Team (2016). R: A language and environment for statistical computing. R Foundation for Statistical Computing, Vienna, Austria. https://www.R-project.org/

**Conflicts of Interest:** The authors declare no conflict of interest.

## Abbreviations

The following abbreviations are used in this manuscript:

| | |
|---|---|
| ECMWF | European Centre for Medium-Range Weather Forecasts |
| m.a.s.l | meters above sea level |
| NCEP | National Centers for Environmental Prediction |
| NOAA | National Oceanic and Atmospheric Administration |
| OI | Optimum Interpolation |
| PrepBUFR | Prepared (Qual. Control.) Binary Universal Format for the Representation of data |
| RRTMG | Scheme selected in WRF for radiation computations (Rapid Radiative Transfer Model) |
| SAR | synthetic aperture radar |
| SST | sea surface temperature (K) |
| WRF | Weather Research and Forecasting Model |
| WRFDA | WRF Data Assimilation System |
| 3DVAR | three-dimensional variational analysis method |
| WSM5 | WRF Single Moment 5 class microphysics scheme |
| $AEP$ | annual energy production (GWh) |
| $CF$ | capacity factor (%) |
| $COE$ | cost of energy (US$) |
| $e$ | partial pressure of water vapour (Pa) |
| $MAPE$ | mean absolute percentage error (%) |
| $P_R$ | rated power of the wind turbine (kW) |
| $P$ | pressure (Pa) |
| $RMSE$ | root mean square error |
| $SCF$ | seasonal capacity factor (%) |
| $SD$ | standard deviation |
| $SEP$ | seasonal energy production (GWh) |
| $T, T_v$ | temperature, virtual temperature (K) |
| $U$ | wind velocity (m/s) |
| $U_n$ | normalized wind velocity (m/s) |
| $WPD$ | wind power density (W/m$^2$) |
| $WPDr$ | wind power density relative error (%) |
| $\rho; \rho_d, \rho_m$ and $\rho_w$ | air density; dry, moist air density and water vapor density (kg m$^{-3}$) |
| $\rho_0$ | standard air density (1.225 kg m$^{-3}$) |

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
