# Peer review of "Seasonal Correction of Offshore Wind Energy Potential due to Air Density: Case of the Iberian Peninsula"

_sustainability, doi:10.3390/su11133648_

Reviewer 1 Report

The paper is of very good quality. Some minor changes are proposed to improve it.

- check the format and avoid pages interruption;

- it is not clear in the methodology where you deal with model verification. I guess it is on paragraph 2.2.3. If this is true, in that case I would suggest to change the paragraph name into "WRDFA model results verification with experimental data"

- check that the nomenclature contains all the symbols cited in the equations

- check further the English language.

Author Response

Thank you for your valuable comments. Following are our responses to your comments:

The paper is of very good quality. Some minor changes are proposed to improve it.
Thanks for your kind words.

- check the format and avoid pages interruption;

Changed, thanks. Take into account that the paper is written in Latex.

- it is not clear in the methodology where you deal with model verification. I guess it is
on paragraph 2.2.3. If this is true, in that case I would suggest to change the
paragraph name into "WRDFA model results verification with experimental data"
Thanks, changed.

- check that the nomenclature contains all the symbols cited in the equations
Checked, thanks.

- check further the English language.
The text has been checked by English Language Services.

Reviewer 2 Report

GENERAL COMMENTS

The work is interesting, it seems to have a clear degree of originality, and it might be appropriate for publication in the journal, after performing a revision.

From this perspective, several changes should be required.

These include:

- Corrections to the figures;

- A careful correction of the English language and rephrasing some paragraphs in order to express in a more comprehensive way the ideas as well as in shorter and clearer sentences.

Some specific comments are given next.

SPECIFIC COMMENTS

TITLE

This is too long and rather confusing, please try reformulate

THE ENGLISH LANGUAGE

This is in general OK in the sense that there are not major mistakes.  Nevertheless, an additional grammar and spelling check would be beneficial. Furthermore, the general clarity of the work should be considerably improved.

KEYWORDS

The first keyword is too long and it should be removed and eventually replaced with two different keywords

ABSTRACT

The abstract is too short. An additional phrase can be added at the beginning concerning the general background.

ABBREVIATIONS

Please, check carefully if all the abbreviations and notations considered in the work are explained for the first time when they are used, even if these are considered trivial by the authors. This is because the paper should be accessible to a wide audience. For example, the significance of m.a.s.l. is provided only in the index and not in the text. Please order in alphabetical order all the abbreviations provided in the index.

SYMBOLS AND EQUATIONS

There are 10 equations presented in the paper and they seem to be OK.

However, some equations are real trivial. So please check if all equations are indeed necessary and also whether all quantities involved are properly defined.

FIGURES & TABLES

Some observations can be made in relationship with the figures, as follows:

Figure 1 – please write on the figure axes the quantity represented and not only the units;

Figure 2 – please write the latitude and the longitude on the figure axes, please modify the Figure captions to: WRF simulation domain and the positions of the six buoys

Figure 4 – please write on the figure axes the quantity represented and not only the units; please denote the two subplots as a) and b) and comment each of them separately in the figure caption.

Figure 5 – please refer to each subplot specifically in the figure caption. Please write on the figure axes the quantity represented as well as the units;

Figure 6 – please refer to each subplot specifically in the figure caption. Please write on the figure axes the units and specify what the green area represents;

Figure 7 – please refer to each subplot specifically in the figure caption. Please write on the figure axes the quantity represented as well as the units;

Figure 8 – please refer to each subplot specifically in the figure caption. Please write on the figure axes the quantity represented as well as the units; Please write also on the colorbar the quantity represented and its units. Since all the colorbars are identical you can provide only one larger colorbar for all subplots.

Figure 9 – please refer to each subplot specifically in the figure caption. Please write on the figure axes the quantity represented as well as the units; Please write also on the colorbar the quantity represented and its units. Since all the colorbars are identical you can provide only one larger colorbar for all subplots.                                           

Table 1 – please write the units for the longitude and latitude. Furthermore the short form for longitude is long

The quantities presented in Table 2 are not explained or given in the index (ex: RRTMG, PREPBUFR)

PAPER STRUCTURE

The introduction should be extended, presenting some similar advances in the specific topic and a more developed literature review. I would suggest moving the description of the methodology and the formula somewhere in the second section.

The sections of Discussions and the Conclusions are too short. Please elaborate more and extend these two sections.  

Author Response

Please, find attached all the answers to reviewers
